# How to Select Patients for Left Ventricular Assist Devices? A Guide for Clinical Practice

**DOI:** 10.3390/jcm12165216

**Published:** 2023-08-10

**Authors:** Daniele Masarone, Brian Houston, Luigi Falco, Maria L. Martucci, Dario Catapano, Fabio Valente, Rita Gravino, Carla Contaldi, Andrea Petraio, Marisa De Feo, Ryan J. Tedford, Giuseppe Pacileo

**Affiliations:** 1Heart Failure Unit, Department of Cardiology, AORN Dei Colli-Monaldi Hospital, 84121 Naples, Italy; 2Department of Medicine, Division of Cardiology, Medical University of South Carolina, Charleston, SC 158155, USAtedfordr@musc.edu (R.J.T.); 3Heart Transplant Unit, Department of Cardiac Surgery and Transplant, AORN Dei Colli-Monaldi Hospital, 84121 Naples, Italy; 4Cardiac Surgery Unit, Department of Cardiac Surgery and Transplant, AORN Dei Colli-Monaldi Hospital, 84121 Naples, Italy

**Keywords:** advanced heart failure, left ventricular assist device, patient’s selection, red flags, golden window

## Abstract

In recent years, a significant improvement in left ventricular assist device (LVAD) technology has occurred, and the continuous-flow devices currently used can last more than 10 years in a patient. Current studies report that the 5-year survival rate after LVAD implantation approaches that after a heart transplant. However, the outcome is influenced by the correct selection of the patients, as well as the choice of the optimal time for implantation. This review summarizes the indications, the red flags for prompt initiation of LVAD evaluation, and the principles for appropriate patient screening.

## 1. Introduction

Management options for patients with advanced heart failure with reduced ejection fraction (advHFrEF) include orthotopic heart transplantation [1,2,3], long-term mechanical circulatory support [4], and intravenous inotropes [5]. Durable mechanical circulatory support through a left ventricular assist device (LVAD) has recently experienced significant technical and scientific development with the advent of fully magnetically levitated centrifugal pump technology. Recent studies indicate that 5-year survival after LVAD implantation approaches that after a heart transplant (5-year survival of 72.5% after heart transplant vs. 58.4 after LVAD implantation) [6]. However, enthusiasm resulting from improved outcomes after LVAD implantation has been tempered by postimplant complications, including stroke [7], infection [8], device thrombosis [9], and right ventricular failure (RVF) [10]. Many of these risks may be mitigated or at least anticipated by the careful selection of candidate patients for LVAD implantation. To provide guidance for general cardiologists, not only advHF specialists, in the proper selection of patients for LVADs, in this review, we provide an overview of the evaluation of advHFrEF patients for LVAD implantation.

## 2. Indications for LVADs

Generally, consideration of LVAD implantation is indicated in all patients with persistent NYHA class III–IV despite guideline-directed medical therapy (GDMT) and poor medium-term prognoses (Table 1) [11]. However, the Interagency Registry for Mechanically Assisted Circulatory Support (INTERMACS) has further classified NYHA III–IV patients with AdvHFrEF [12]. According to this classification, the best outcome after LVAD implantation is achieved in INTERMACS class 2–3 patients [13]. However, recent technological advances have reduced postoperative complications and have paved the way for “early” LVAD implantation, which is now also being considered for patients in INTERMACS class 4 [14] (Figure 1).

There are three main indications for LVAD implantation:►Bridge to Transplant (BTT): Indicated in patients on the orthotopic heart transplant list unable to maintain adequate end-organ perfusion with GDMT alone;►Bridge to Candidacy (BTC): Suitable in selected patients with temporary contraindications to orthotopic heart transplantation that could be reversed after support with an LVAD (e.g., increased pulmonary vascular resistance, functional kidney disease);►Destination Therapy (DT): Recommended in patients with an absolute and permanent contraindication to orthotopic heart transplantation, in whom an LVAD is used as a long-term therapy.

## 3. Clues to Prompt Initiation of LVAD Evaluation

Deriving the maximal mortality and morbidity reduction from LVAD implantation depends on the timing of the implant [15]. Proceeding with LVAD implantation too early in a patient’s heart failure course risks obviating the known benefit of pharmacologic and device guideline-directed therapy while exposing the patient to unnecessary perioperative risk. In contrast, if the LVAD is implanted too late in the course of the disease, the postoperative prognosis may be compromised by multiorgan dysfunction due to prolonged advHFrEF status [16]. Some degree of progressive multiorgan dysfunction is nearly ubiquitous in patients with end-stage heart failure and must be carefully evaluated when selecting candidate patients for LVADs. Therefore, identifying and appropriately referring patients with advHFrEF to centers capable of advanced therapies is critical to performing LVAD implantation in the “golden window” and, thus, to maximizing benefits and minimizing surgical procedure risks [17]. To help identify patients who may benefit from an LVAD at an appropriate time, several clinical, laboratory, and echocardiographic markers have been identified as “red flags”, indicating that the conventional approach is not enough for the patient and that LVAD implantation should be considered (Table 2).

### 3.1. Clinical and Laboratory Red Flags

Two or more episodes of *worsening heart failure* in the last 12 months indicates a poor prognosis in patients with advHFrEF [18]. A retrospective analysis involving 14,374 patients hospitalized for HF showed that patients with two or more hospitalizations for HF had a 1-year mortality rate of >40% [19]. More recently, a Swedish registry that enrolled 1801 patients with HF showed that worsening HF (defined as hospitalization for HF or an urgent need for intravenous diuretics) is associated with a reduction in the 5-year survival rate, particularly in patients with a left ventricular ejection fraction of ≤25% (log-rank *p* < 0.0001) [20].

The onset of *intolerance to disease-modifier* drugs due to symptomatic hypotension, resulting in the reduction or discontinuation of guideline-directed medical therapy (GDMT), is associated with a poor prognosis [21] unless it is due to hypovolemia [22]. In a retrospective analysis including 259 adult patients discharged from hospital after an episode of worsening heart failure, it was shown that those without an angiotensin-converting enzyme inhibitor because of hypotension or renal intolerance had a mortality rate higher than that of patients on angiotensin-converting enzyme inhibitors (57% vs. 22%; *p* = 0.0001) [23].

*Inotrope dependence* (the onset of symptomatic hypotension, recurrent congestive symptoms, or worsening renal function when inotropes are withdrawn) is a clear negative prognostic marker in patients with worsening advHFrEF [24] and should prompt the consideration of advanced therapies, including LVADs and/or heart transplant.

*A high diuretic dose* or the need to *escalate the diuretic* dose is also associated with poor prognosis. A hallmark study that enrolled 1354 patients with advHFrEF showed that patients in the highest quartile for diuretic dose (i.e., ≥160 mg of furosemide/daily) had the worst prognosis [25].

*Arrhythmic storms* (>3 episodes of sustained ventricular tachycardia or ventricular fibrillation in 24 h) are associated with a poor prognosis, mainly if they occur in patients in New York Heart Association (NYHA) functional class III–IV [26] or in a patient with prior ablation of the arrhythmic substrate [27]. Careful consideration of the arrhythmic substrate is required during evaluation, as ventricular tachyarrhythmias after LVAD implantation may also pose considerable risk for poor outcomes.

Numerous studies have shown that *hyponatremia* is a marker of poor prognosis in patients with acute heart failure [28,29]. More recently, the prognostic importance of persistent hyponatremia was documented in patients with advHFrEF in the ESCAPE (Evaluation Study of Congestive Heart Failure and Pulmonary Artery Catheterization Effectiveness) trial. In fact, in this study, the persistence of hyponatremia was associated with increased all-cause mortality and hospitalizations (73% vs. 50%; HR, 1.54) when compared with normonatremic patients [30].

The persistence of *elevated plasma levels of NT-proBNP* despite GDMT is associated with poor prognosis in patients with advHFrEF. In a prospective study that enrolled 550 patients with heart failure of ischemic and non-ischemic etiology, NT-proBNP levels of >5000 pg/mL were associated with increased mortality (mortality rate of 28.4% per year), which markedly discriminated advHFrEF candidates for LVAD implantation or urgent heart transplantation [31]. Furthermore, in the BioSHiFT (Role of Biomarkers and Echocardiography in Prediction of Prognosis of Chronic HF Patients) study, an increase in serial assessments of plasma NT-proBNP levels was associated with cardiac death, heart transplantation, or LVAD implantation [32].

### 3.2. Echocardiographic Red Flags

Certain echocardiographic features may suggest the presence of an end-stage form of heart failure, such as the following [33]:►Marked left ventricular dilatation (left ventricular end-diastolic diameter of >80 mm);►Left ventricular ejection fraction of ≤25%;►Persistent restrictive mitral filling patterns or echocardiographic findings suggesting pulmonary hypertension despite aggressive diuretic therapy and therapy with inodilators (i.e., milrinone, levosimendan) [34].

Despite the importance of echocardiographic red flags, they should always be contextualized in the clinical scenario to best determine whether a patient warrants referral for the consideration of LVADs or other advanced therapies.

### 3.3. Invasive Hemodynamics Red Flags

Right heart catheterization is often used in patients with suspected advHFrEF to quantify the degree of hemodynamic dysfunction [35].

The persistence of a reduced cardiac index or increased left ventricular filling pressures despite the aggressive use of intravenous diuretics and inodilators is an important predictor of poor outcomes [36] and warrants evaluation for an LVAD.

## 4. Assessment of Patients

Although the evaluation of patients for LVADs should be performed in specialized centers for advHFrEF treatment, we consider it useful for the clinical cardiologist to be familiar with the basic principles of LVAD preimplant evaluation. Therefore, in this session, we practically cover the rationale for the primary investigations that a patient candidate for LVAD implantation undergoes in the preoperative phase. The class of recommendations (COR) and level of evidence (LOE) reported for each single indication are provided according to the latest recommendations by the International Society for Heart and Lung Transplantation [15].

### 4.1. Blood Investigation

Blood analysis represents an important step for the evaluation of patients with advHFrEF as potential candidates for LVAD implantation; mainly, the following parameters should be evaluated:►*Kidney function:* Because the presence of advanced chronic kidney disease (CKD) strongly affects the LVAD post implantation prognosis [37], evaluation of serum creatinine blood urea nitrogen, creatinine clearance, and proteinuria is recommended in all patients (COR I, LOE C). Such assessments should be done in hemodynamically stable patients or after an adequate period of stabilization with diuretics, inotropes, and/or temporary mechanical circulatory support, if clinically indicated (COR I, LOE C). In patients with CKD status >IV–V (i.e., an estimated glomerular filtration rate of <30 mL/min/1.73 m^2^), the implantation of an LVAD should also be carefully evaluated by a multidisciplinary team (COR IIa, LOE C) as the anticipation of permanent dialysis is a contraindication to LVAD implantation (COR III, LOE C).►*Hepatic function:* Severe hepatic dysfunction is strongly related to an adverse prognosis post-LVAD implantation [38], so a liver enzyme and hepatic protein synthesis evaluation is mandatory in all patients.►*Blood glucose and glycated hemoglobin:* Diabetes has been associated with infection and late mortality in LVAD patients [39], so in all LVAD candidates, the presence of diabetes should be carefully evaluated (COR I, LOE C). Patients with uncontrolled diabetes should undergo optimization of their glycemic control (endocrine expertise recommended; COR I, LOE C), and patients with diabetes-related organ damage (proliferative retinopathy, vasculopathy, nephropathy) have a relative contraindication for LVAD implantation (COR IIb, LOE C).►*Platelet count and coagulation parameters:* Because major post-LVAD complications include both thrombotic (particularly pump thrombosis) and hemorrhagic (mainly gastrointestinal) events [40,41], all patients should undergo preoperative evaluation of their international normalized ratio of prothrombin time, activated prothrombin time, and platelet count (COR I, LOE C). Baseline abnormalities in coagulation parameters not due to drug therapy or an altered platelet count (thrombocythemia and thrombocytosis) should be promptly evaluated to determine their etiology before LVAD implantation (COR I, LOE C).

### 4.2. Echocardiography

All patients should undergo echocardiography before LVAD implantation to assess their right ventricular size and function and to identify and quantify any coexisting valvulopathies (COR I, LOE C).

►*Right ventricle evaluation*: Most echocardiography-based predictors of right ventricular failure (RVF) post-LVAD implantation are based on assessing RV systolic–diastolic function. However, it was recently demonstrated that evaluating the RV–pulmonary artery (PA) parameters may increase the possibility of predicting postimplant RVF [42]. A systolic tricuspid annulus plane excursion (TAPSE) of <8 mm has been associated with RVF [43,44]; similarly, a right ventricular free wall s′ value on tissue Doppler of <5–8 cm/s [45] and an RV fractional area change (FAC) value of <25–30% [45] are associated with postimplant RVF. RV diastolic dysfunction also provides essential information on the occurrence of RVF after LVAD implantation. Indeed, a preimplant tricuspid E/e′ ratio of >10 predicts RVF [46]. Finally, recently, measures of the RV–PA coupling, which are less affected by afterload changes [47], have been shown to predict the onset of post-LVAD RHF. Specifically, a value of <24 obtained from the product of the peak systolic longitudinal strain rate of RV and the mean RV–PA gradient [48] or an RV load adaptation index of <14 [49] identifies patients with a failing RV unable to adapt to the increase in blood flow post-LVAD implantation.►*Valvular evaluation:* Aortic valve regurgitation results in recirculation and reduced forward flow in patients with LVADs [50]. Therefore, the presence of more than mild aortic valve regurgitation is an indication for valve replacement with a biological valve (COR I, LOE C). Conversely, mild to moderate aortic stenosis does not impact LVAD function unless there is concomitant valve insufficiency [51]; therefore, it is common practice to replace the aortic valve only in patients with pre-existing severe aortic stenosis (COR I, LOE C). Significant mitral stenosis prevents inflow to the LVAD [52]; therefore, moderate or more significant mitral stenosis indicates the need for commissurotomy or valve replacement with a biological valve (COR I, LOE C). Mitral insufficiency, even of severe grade, is not a contraindication because left ventricular unloading almost always reduces the severity of mitral regurgitation [53,54]; on the other hand, since significant tricuspid regurgitation contributes to the occurrence of RVF [55,56], repair or replacement with a bioprosthetic valve should be performed in patients with a moderate or greater degree of tricuspid valve insufficiency that is anticipated to persist after LVAD support (COR IIb, LOE B).

Recommendations for patients with preexisting prosthetic valves are shown in Table 3.

Concomitant valve surgery with LVAD implantation increases the surgical risk due to the prolonged extracorporeal circulation time and increased incidence of postoperative RVF [57]. These factors must be considered during the patient selection process.

### 4.3. Other Imaging Techniques

In addition to transthoracic echocardiography, other imaging techniques should be used for the appropriate selection of candidate patients for LVAD implantation:►*Chest imaging*: Evaluation of the intrathoracic anatomy is mandated in the preoperative assessment for LVAD surgery [58]. Therefore, chest radiography should be performed in all patients (COR I, LOE C). In patients with suspected intrathoracic abnormalities or previous cardiac surgery, computed tomography (CT) or magnetic resonance imaging should be used to better define the thoracic anatomy (COR I, LOE C).►*Liver imaging*: In patients whose clinical history and/or laboratory investigations suggest hepatic dysfunction, screening for fibrosis or cirrhosis via ultrasonography or CT scan (COR I, LOE C) is necessary. Since fibrosis in the absence of significant portal hypertension may not be a contraindication to LVAD implantation [59], especially in the case of a normal hepatic synthetic capacity, a consultation with a hepatologist is indicated in such patients (COR I, LOE C). Conversely, cirrhosis or advanced hepatic dysfunction represents a contraindication to LVADs. A biopsy may be required to assess for cirrhosis. An analysis of 524 advHFrEF patients undergoing LVAD implantation showed that patients who had a MELDIX (End-Stage Liver Disease eXcluding International Normalized Ratio model) score of >14 had poorer short- and long-term survival, as well as an increased risk of early right ventricular dysfunction [60]. For this reason, patients with confirmed cirrhosis or an increased MELD score of >14 are not eligible for an LVAD (COR II, LOE B).►*Gastrointestinal imaging*: Gastrointestinal bleeding is present in 15–30% of LVAD patients; the rate is higher in patients with previous gastrointestinal bleeding or colon polyposis [61]. Therefore, in patients with precedent gastrointestinal bleeding or with suspected unexplained iron deficiency anemia [62], screening with upper and lower endoscopy should be considered (COR IIa, LOE C).

### 4.4. Invasive Hemodynamics

An invasive hemodynamics assessment by right heart catheterization should be performed in all patients before LVAD implantation (COR I, LOE C) to measure their intracardiac filling pressures and assess their RV function [63]. The main hemodynamic parameters that predict RVF after LVAD implantation are shown in Table 4. In patients in whom these parameters indicate a high risk of RVF, optimization with diuretics, vasopressors/inotropes, dialysis, or temporary percutaneous mechanical circulatory support should be performed (COR I, LOE C). After optimization, if the RV function remains suboptimal, biventricular assist device support should be considered [64].

### 4.5. Psychosocial Evaluation

All patients who are candidates for LVADs should undergo a comprehensive psychosocial assessment (COR I, LOE C) to identify the following:►Nonadherence to drug therapy;►Unsafe home environment;►Absence of social support;►Presence of mental illness;►Presence of drug or alcohol dependence.

All the above factors are associated with worse outcomes after LVAD implantation, so the candidacy for an LVAD of a patient with one or more of these factors should be carefully considered by a multidisciplinary team (cardiologist, cardiac surgeon, anesthesiologist, psychologist/psychiatrist, internist) [65,66].

## 5. Conclusions

LVAD implantation represents a valuable therapeutic option in carefully selected patients with advHFrEF. Technological innovations and improved implantation techniques ensure an immediate improvement in symptoms, quality of life, and short- and medium-term survival. However, the high frequency of postimplant adverse events and the high economic and resource costs are still barriers to adopting this treatment for a broader population of patients with advHFrEF. Therefore, improving the patient selection process is crucial to making LVAD implantation an acceptable and cost-effective alternative to orthotopic heart transplantation.

## Figures and Tables

**Figure 1 jcm-12-05216-f001:**
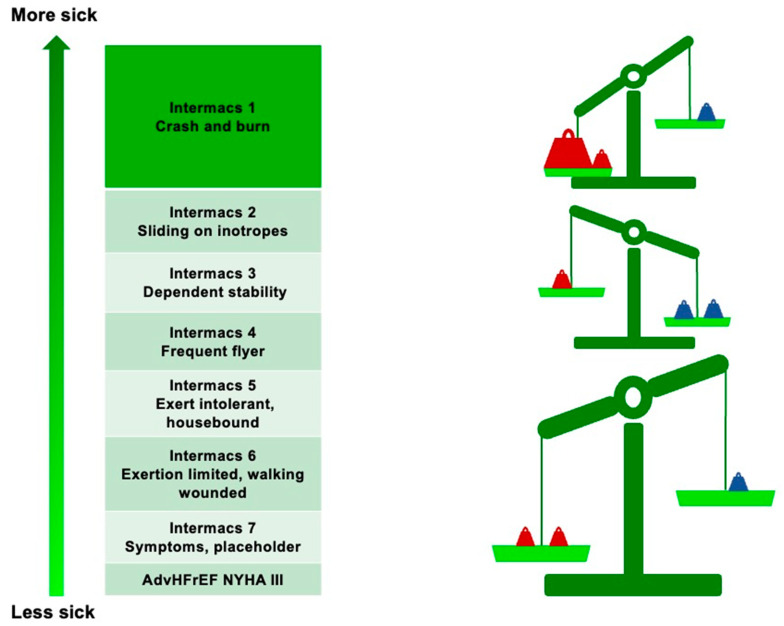
Risk/benefit ratio of LVAD implantation according to INTERMACS class. Red weights represent the risks; blue weights represent the benefits.

**Table 1 jcm-12-05216-t001:** Indications to LVADs according to international guidelines. ISHLT: International Society Heart and Lung Transplantation; AdvHFrEF: advanced heart failure with reduced ejection fraction; NYHA: New York Heart Association; GDMT: guideline-directed medical therapy; ACC/AHA: American College of Cardiology/American Heart Association; LVEF: left ventricular ejection fraction; CI: cardiac index; PCWP: pulmonary capillary wedge pressure; EACTS: European Association for Cardiothoracic Surgery; INTERMACS: Interagency Registry for Mechanically Assisted Circulatory Support; CPET: cardiopulmonary exercise test, COR: Class of Recommendation, LOE: Level of Evidence, NA: Not Available.

**ISHLT (2023)**		
Indications	COR	LOE
As a bridge to transplant AdvHFrEF patients with severe symptoms (NYHA functional class IIIB–IV) refractory to maximal medical management, who are inotrope dependent or on temporary mechanical circulatory support if transplant is unlikely to occur in the short term	I	A
As destination therapy AdvHFrEF patients ineligible for heart transplant with severe symptoms (NYHA functional class IIIB–IV) refractory to maximal medical management, who are inotrope dependent or on temporary mechanical circulatory support	I	A
**ACC/AHA guidelines (2022)**		
AdvHFrEF patients with NYHA functional class IV symptoms despite GDMT and device therapy who are deemed to be dependent on inotropes or mechanical circulatory support	I	A
AdvHFrEF with NYHA functional class IV symptoms despite GDMT	IIa	B
**ESC Guidelines (2021)**		
Indications	COR	LOE
AdvHFrEF patients with severe symptoms despite GDMT and device therapy who have at least one of the following: - LVEF < 25%, peak VO2 < 12 mL/kg/min, and/or <50% predicted value - More than three HF-related hospitalizations - Inotrope dependence - Mechanical circulatory support dependence - Worsening renal and/or hepatic function or type II pulmonary hypertension due to reduced perfusion (CI < 2 L/min with PCWP > 20 mmHg)	NA	NA
**EACTS (2019)**		
As destination therapy/bridge to transplant AdvHFrEF in NYHA functional class IIIB–IV with LVEF < 25% and at least one of the following criteria: - INTERMACS class 2–4 - Hemodynamic dependence by inotrope or short-term mechanical circulatory support - Progressive end-organ dysfunction - Peak VO2 < 12 mL/kg/min at CPET	IIa	B
As a bridge to candidacy AdvHFrEF in NYHA functional class IIIB–IV with LVEF < 25% and - Elevated pulmonary vascular resistance or potentially reversible renal failure Recent cancer, obesity, or recovering drug or alcohol dependence in potential heart transplant candidates	IIb	B

NA: Not Available.

**Table 2 jcm-12-05216-t002:** Red flags that indicate the need to consider an LVAD implant.

**Clinical**
Two or more episodes of worsening heart failure in the last 12 months
Down-titration of guideline-directed medical therapy due to hypotension, dizziness, or excessive fatigue
Need for a high diuretic dose (e.g., >160 mg/d furosemide)
Inotrope dependence
Arrhythmic storms
**Laboratory**
Persistent hyponatremia
Elevated plasma levels of NT-pro brain natriuretic peptides (e.g., >5000 pg/mL)
**Echocardiography**
Left ventricular end-diastolic diameter ≥ 80 mm
Left ventricular ejection fraction ≤ 25%
Restrictive filling pattern and/or pulmonary hypertension despite diuretics and inodilators
Right ventricular dysfunction
**Invasive hemodynamics**
Low cardiac output and high filling pressure despite diuretics and inodilators

**Table 3 jcm-12-05216-t003:** Surgical recommendations for LVAD candidate patients with prosthetic valves. LVAD: left ventricular assist device.

Type of Prosthetic Valve	Recommendation	Note
Functioning biological prostheses (regardless of the anatomical site)	No removal or replacement at the time of implant (COR I, LOE C)	A biological valve, whether in the aortic or mitral position, is well tolerated during LVAD support
Mechanical aortic valve	Replacement with a bioprosthetic valve during LVAD implantation (COR I, LOE B) or path closure when no other options are feasible (COR IIb, LOE C)	Mechanical aortic valves may result in thromboembolic events due to blood stasis around the valve and intermittent valve opening
Mechanical mitral valve	Replacement of a properly functioning mechanical mitral valve is not recommended (COR III, LOE C)	Exchanging a mechanical mitral valve is technically very complex

**Table 4 jcm-12-05216-t004:** Hemodynamics-based parameters associated with right ventricular dysfunction after LVAD implantation. CVP: central venous pressure; PCWP: pulmonary capillary wedge pressure; MAP: mean arterial pressure; PAPi: pulmonary artery pulsatility index; RVSWI: right ventricular stroke work index; SVI: stroke volume index; PACI: pulmonary artery compliance index.

Hemodynamic Parameter	Cut-Off Associated with RVF
Preoperative CVP	>10 mmHg
CVP/PCWP	>0.63
MAP/CVP	<7.5
PAPi	<1.85
RVSWI	<0.30 mmHg/L/m^2^
SVI	<22.1 mL/m^2^ during systemic vasodilator drug challenge
PACI	<0.89 mL/mmHg/m^2^

## Data Availability

No data were generated in this study.

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
