# Peer review of "How to Select Patients for Left Ventricular Assist Devices? A Guide for Clinical Practice"

_jcm, 2023, doi:10.3390/jcm12165216_

Round 1

Reviewer 1 Report

Masarone et al put forth a great manuscript review regarding the selection of patients for left ventricular assist device. The manuscript is thorough and succinct and provides an excellent overview of what can sometimes be a difficult and complex topic. It will be a great contribution to the field. 

I have a few minor suggestions:

1) While I do not disagree that the HM3 VAD is a major innovation with significant improvements in outcomes, I do believe that stating that the 5 year survival after LVAD approaches that of heart transplant is somewhat misleading without the actual numbers. If the authors want to make this statement, I would include the actual latest specific data ( ISHLT data for 5 y OHT survival ~ 72.5%, 5 year HM3 survival data 58.4%). 

2) Page 1, line 27, "advanced heart failure reduced ejection fraction" should read "advanced heart failure WITH reduced ejection fraction"

3) Page 4, line 74-75: "unnecessary perioperative" I assume should read "unnecessary perioperative risk"

4) Page 4, line 84-85: pharmacological and device based, would consider clarifying what device based is referring to since LVADs are technically devices (I assume the authors mean ICD, CRT, mitral clip, etc but not mechanical circ support). 

5) Table 4: correction to PAPI, PAPI = (PASP - PADP)/CVP, so PAPI already includes CVP. Would change to just PAPI (without / CVP) or otherwise detail the full equation. 

Author Response

Masarone et al put forth a great manuscript review regarding the selection of patients for left ventricular assist device. The manuscript is thorough and succinct and provides an excellent overview of what can sometimes be a difficult and complex topic. It will be a great contribution to the field. 

 Response: Thank you to the reviewer for their appreciation of our work

I have a few minor suggestions:

1) While I do not disagree that the HM3 VAD is a major innovation with significant improvements in outcomes, I do believe that stating that the 5 year survival after LVAD approaches that of heart transplant is somewhat misleading without the actual numbers. If the authors want to make this statement, I would include the actual latest specific data ( ISHLT data for 5 y OHT survival ~ 72.5%, 5 year HM3 survival data 58.4%). 

2) Page 1, line 27, "advanced heart failure reduced ejection fraction" should read "advanced heart failure WITH reduced ejection fraction"

3) Page 4, line 74-75: "unnecessary perioperative" I assume should read "unnecessary perioperative risk"

4) Page 4, line 84-85: pharmacological and device based, would consider clarifying what device based is referring to since LVADs are technically devices (I assume the authors mean ICD, CRT, mitral clip, etc but not mechanical circ support). 

5) Table 4: correction to PAPI, PAPI = (PASP - PADP)/CVP, so PAPI already includes CVP. Would change to just PAPI (without / CVP) or otherwise detail the full equation. 

 Response: Thank you to reviewer for these useful comments we have corrected the text accordingly

Reviewer 2 Report

Indeed choosing the correct population for LVAD transplantation is a crucial step ensuring immediate improvement in symptoms, quality of life, and short- and medium- term survival of our patient. Post-implant complications are a lot including stroke, infection, device thrombosis, and right ventricular failure so in  that way we have to be very carefull.

Good clinical aproach of the subject, fully scientifically informed. There could be better figures, and tables

Author Response

Indeed choosing the correct population for LVAD transplantation is a crucial step ensuring immediate improvement in symptoms, quality of life, and short- and medium- term survival of our patient. Post-implant complications are a lot including stroke, infection, device thrombosis, and right ventricular failure so in  that way we have to be very carefull.

Good clinical aproach of the subject, fully scientifically informed. There could be better figures, and tables

Response: Thank you to the reviewer for the appreciation of our work